# Adoption Case of IIoT and Machine Learning to Improve Energy Consumption at a Process Manufacturing Firm, under Industry 5.0 Model

Andrés Redchuk [1] , Federico Walas Mateo [2,3,*] , Guadalupe Pascal [2] and Julian Eloy Tornillo [2]

1 Computer Science and Engineering Department, Universidad Rey Juan Carlos, 28933 Madrid, Spain
2 Engineering Faculty, Universidad Nacional de Lomas de Zamora, Lomas de Zamora 8659, Argentina
3 Engineering and Agronomy Institute, Universidad Nacional Arturo Jauretche, Florencio Varela 1888, Argentina
* Correspondence: fwalas@unaj.edu.ar or fedewalas@gmail.com

**Abstract:** Considering the novel concept of Industry 5.0 model, where sustainability is aimed together with integration in the value chain and centrality of people in the production environment, this article focuses on a case where energy efficiency is achieved. The work presents a food industry case where a low-code AI platform was adopted to improve the efficiency and lower environmental footprint impact of its operations. The paper describes the adoption process of the solution integrated with an IIoT architecture that generates data to achieve process optimization. The case shows how a low-code AI platform can ease energy efficiency, considering people in the process, empowering them, and giving a central role in the improvement opportunity. The paper includes a conceptual framework on issues related to Industry 5.0 model, the food industry, IIoT, and machine learning. The adoption case's relevancy is marked by how the business model looks to democratize artificial intelligence in industrial firms. The proposed model delivers value to ease traditional industries to obtain better operational results and contribute to a better use of resources. Finally, the work intends to go through opportunities that arise around artificial intelligence as a driver for new business and operating models considering the role of people in the process. By empowering industrial engineers with data driven solutions, organizations can ensure that their domain expertise can be applied to data insights to achieve better outcomes.

**Keywords:** IIoT; energy efficiency; machine learning; low-code platform; United Nations Sustainable Development Goals; Industry 5.0

## 1. Introduction

In this work, we address the adoption case of machine learning (ML) through a low-code platform (LCP) [1] in a food process company to optimize energy consumption. The article goes further from previous research considering the emerging Industry 5.0 (I5.0) paradigm with its human-centered and environment sustainability vision.

Some authors [2–6] sheds light on the scope and objectives of the I5.0 model, where sustainability is aimed at integration in the value chain and centrality of people in the production environment. Moreover, the European Commission [7,8] observes that I5.0 is complementary to Industry 4.0 paradigm, driving industries to a sustainable, human-focused, and resilient industry. Finally, both articles highlight that the 5.0 model moves the scope from solely shareholder value to community value.

The article from [9] highlights AI's and ML's disruptive potential, opening opportunities for new business models and entrepreneurs, lights the disruptive potential of AI, and ML, opening opportunities for new business models and entrepreneurs. The paper refers to conceptual development with many cases from practice. It concludes that the innovative potential of the new business models opens the door to radical new operating models that could lead to new technological firms. On this line, the article sheds light on

the contributions that new business and operating models on ML can handle to improve industrial processes.

Nowadays, process industries are focused on energy efficiency to improve environmental sustainability in the Net Zero Carbon emissions objective by reducing energy consumption. The article [10] considers the case of the food industry, which requires process heat, most of which was supplied by fossil-fuel-based technologies in 2019. A significant increase in the electrification of process heat generation is assumed to occur. To achieve the overall $CO_2$ emissions targets, electricity generation will increase the average global renewable electricity share from 25% in 2019 to 74% in 2030. That means significant efforts should be made to make traditional heating in food processing more efficient.

The concepts in the above paragraph are reinforced when reviewing the contribution to the United Nations (UN) Sustainable Development Goals (SDGs) [11] in particular, the ones referring to industrial innovation, responsible consumption and production, and climate actions. The work from [12] presents a comprehensive review of AI in forming Sustainable Business Models (SBMs). AI and SBMs are relatively nascent fields of research. The results indicate that the literature focused on only some aspects involved in sustainable development through AI. The paper observes that there is an important gap to reach the guidelines established for companies by the UN 2030 Agenda. To achieve high sustainability standards, it is necessary to improve the technical-scientific quality of the production systems, and AI can represent the vehicle to this improvement.

This article considers the issues in the previous paragraphs regarding opportunities using data models to generate value from them, and introduces a methodology to ease the evolution of traditional production systems into I5.0 paradigm. The focus of the paper is to present a methodology to ease the adoption of AI/ML in industrial environments, and integrate an existing IIoT platform that generate valuable data from industrial operations. The authors intend to illustrate the way the AI/ML platform with a Low-Code solution approach a Lean Startup methodology can produce results in less time, generating a co-creation environment, to optimize an industrial process in a traditional industry.

This paper structure begins with a description of related concepts and current approaches in the literature regarding data collection, IT/OT infrastructure, ML, and data integration in industrial environments toward energy efficiency. The article continues to describe the methodology proposed, and presents the studied case; it is described giving details on how the proposed methodology is applied.

## 2. Conceptual Framework and References in Process Industries

This section presents approaches and concepts relevant to energy optimization in process industries using AI. Besides the solution, implementation process under the I5.0 paradigm is considered.

Regarding smart production in the food industry, current smart manufacturing approaches focus on intelligent data collection and its analysis with ML algorithms [12]. This includes a variety of data sources, including raw material, machine, or market data (e.g., information regarding sales or complaints).

The topic of information collection and integration has been covered by authoritative standardization bodies such as the International Society of Automation (ISA) [13] and the International Electrotechnical Commission (IEC) [14]. As an example, the multi-layer IEC 62264 standard based on the ISA-95 [15] establishes a data model exchange architecture enabling the integration of applications running in business an industrial area of a firm. Enterprises complying with the standard can define interfaces between control and business functions, allowing them to make informed decisions on data to exchange to minimize risks and costs in case of implementation mistakes.

The article [16] highlights the conceptual framework of information technology (IT) and operational technology (OT) infrastructure that enables the I5.0 model. The convergence of OT/IT is critical for the integrating data and AI in the industrial decision-making process, creating the basis for a cognitive plant. The paper includes a real case that fulfills

the specific needs of OT and IT, achieving fast and homogeneous transfer of large volumes of data towards the IT layer.

In line with the article cited below is the work from [17], which present a conceptual framework and preliminary findings to go deeper into a bibliometric analysis around the idea of artificial intelligence and machine learning (AI/ML) as a tool for the optimization of processes within the Industry 4.0 model. The paper develops the role of IIoT to ease the adoption of advanced analytics tools in industrial processes.

Authors of [18] present a case where an electric arc furnace in a steel industry has been optimized through an ML method, leading to lower energy consumption. In the study, different machine learning and data processing methods were used to evaluate the energy efficiency parameters of the furnace process. The authors point out that the dataset was collected over five years, in a steelmaking factory, with 42 features. The article explains the complexities to be solved on how to implement the methodology to optimize an industrial process successfully.

Another paper that gives insights into this study is the work [19], which, through an empirical case study, evaluates the proposed method's effectiveness and efficiency compared to existing ones from the literature on an industrial process. The research highlights the importance of considering domain knowledge in feature selection to build a robust industrial ML model. They define the industrial process as a complex network of thousands of elements interconnected by material, energy, and information flows. Finally, the paper proposes a feature selection method to capture domain knowledge and identify relevant process signals.

The research in article [20] refers to the responsibility of industries to improve energy efficiency and minimize carbon footprints. The authors developed a model-free demand response (DR) scheme for industrial facilities. The model was used in a real case in industry, and generated an optimal energy consumption schedule, downsizing energy costs while production matched to demand.

Finally, the paper [9] presents the case of the adoption of a ML solution in a steel manufacturing process through a low code platform. The paper includes research to visualize the state-of-the-art of AI/ML adoption in steel manufacturing industries to optimize processes. The results of the case highlighted the way the innovative business model, based on a Low-Code solution, achieved results in less time than conventional approaches of analytics solutions. The contribution of this article was the proposal of an innovative methodology to put AI/ML in the hands of process operators. It aimed to show how it was possible to achieve better results in a less complex and time-consuming adoption process. The work also addresses the need for an important quantity of data from the process to successfully adopt this kind of solution.

## 3. Implementation Using a Low Code AI Platform and Lean Startup

The methodology to be applied in the case considers the opportunities to use ML in a food manufacturing firm by using the Low-Code software solution [21], complemented by Lean Startup methodology [22]. The implementation strategy proposed aims to have results faster than traditional methodologies to adopt data driven solutions, making possible to ease the deployment of AI in traditional industrial environments.

The software solution is an LCP offered to industrial companies to add value to the data generated in the operation, and consists of tools for data visualization, and normalization to advance in the ML model. The platform also has preconfigured templates to address different specific problems of industrial processes. The platform offers six templates: Forecasting, Anomaly Detection, Optimization, Simulation, Failure Prediction, and Defective Part Prediction. The template to use in each case depends on the nature of the process and the opportunity to address. Figure 1 shows the preconfigured templates of the LCP and the general use of each approach.

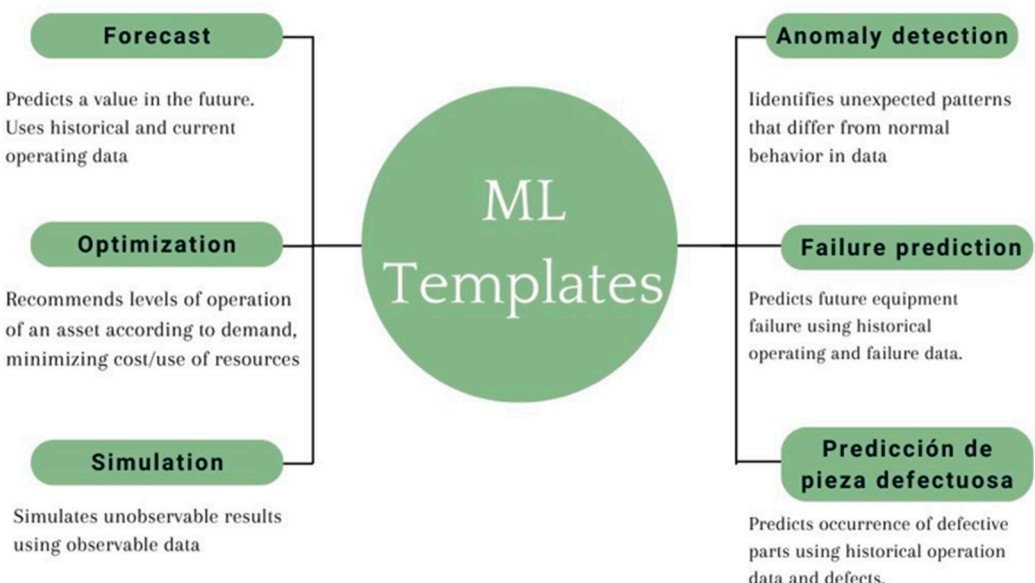

**Figure 1.** Schema of the six preconfigured templates the LCP offer, and the scope of each one. Source: authors.

The templates presented by the software solution are artifacts preconfigured as part of the GUI provided by the platform. This feature makes the user on the production floor, the expert in the industrial process, to understand in a visual way, and therefore in a more user-friendly way, the achievements that are intended to be obtained with the data model. These speeds up the evaluation of the model and makes it easier to generate results early, and thus be able to evaluate the effectiveness of the proposal in advance.

The adoption cycle begins with the analysis of the data, the LCP provides visual tools to analyze the quality and consistency of the data. Through this type of functionality, data is curated, normalized and validated. A data standardization and validation process is developed to eliminate the redundancies or inconsistencies, completing data that updates each record, eliminating anomalies, and reviewing the most relevant values.

The solution is offered in as a service (SaaS) model, ingesting data in a data series format from Industrial Internet of Things (IIoT) platforms. This way the ML solution is linked to the plant with the operations management solution, and then generate the predictions to empower people at the industrial process.

In order to generate a co creation environment among people from the industry and the software vendor staff, a Lean Startup methodology is used. This way non-value adding activities are minimized and people from the industrial process can be involved earlier, while introducing the new solution. The Lean Startup methodology includes three key issues: to experiment instead of planning, ease innovative practices, and agile approach [23].

The experimentation process is described by the Build-Measure-Learn feedback loop consisting of three steps: build, measure, and learn. In the first step, build, it is essential to create a minimum viable product (MVP) using as less resources as possible after identifying the most important hypotheses. The goal of building an MVP is to identify the proposed solution's potential [24] and the value for the user. The measure step aims at collecting data that can verify or dismiss the hypothesis made about the solution to be offered. In the learn step, the goal is to know about the investigated hypotheses from collected data. The learning process shows whether an underlying hypothesis can be verified or not and indicates if the MVP is a viable solution to the customer problem.

Lean Startup methodology is described by a Build-Measure-Learn feedback loop. In the first step, a minimum viable product (MVP) is created using as less resources as possible after identifying the model [23]. The objective of having an MVP is to evaluate the proposed solution's [24] and the value for the user. Then in the following steps the model is trained

and evaluated to verify or dismiss the model. In the last step, the validated process is institutionalized.

Authors of [25] present a case to validate the use of Lean Startup methodology to ease innovation in stablished enterprises. The work observes two key points to recommend the use of the strategy. The first refers to the optimization of the use of resources and achieving results in less time than traditional approaches, and the second issue is the stronger involvement of users involved in the innovation project.

## 4. The Case

A leading food ingredient company production company in North America wanted to optimize its operations across their food production facilities. The project started after defining the use case and its scope. One of the improvement opportunities or "pain Points" identified was making their energy production processes more efficient. Energy was the second cost behind raw material, and the firm has a goal to reduce 25% $CO_2$ emissions by 2030.

The industrial operations team wanted to utilize AI to predict each boiler's thermal efficiency so that the gas loading could be optimized according to efficiency. The process perfectly fits the organization's goals to cut energy consumption, running costs, degradation across the turbine portfolio, and carbon emissions.

The industrial company uses natural gas turbines and boilers to generate power and steam for the plant. Although the boilers were provided by the same supplier, the behavior of each device differs with changing environmental conditions. Each boiler's thermal performance may fluctuate due to different configurations and piping, environmental conditions, and uneven wear and tear, meaning that they were consuming the same natural gas rate but not necessarily producing the same output. Figure 2 shows how the system works.

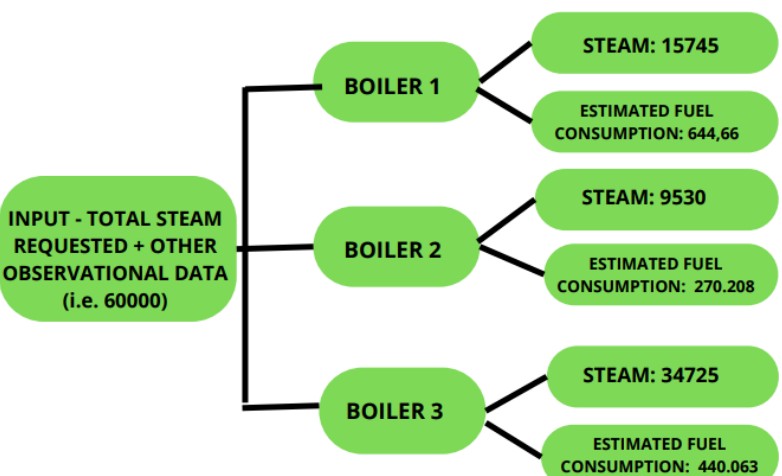

**Figure 2.** Based on the thermal efficiency of each boiler, the problem to solve is to estimate the optimum fuel required for each boiler while ensuring overall steam production meets the demand. Source: Authors.

The challenge was to balance the load among the three boilers producing enough steam to match the demand. The set points were set manually by the operators, depending on their knowledge and experience. Then efficiency was different among production shifts. The adoption of AI could help to predict the valve set points to the boilers in order to optimize the system. Some considerations had been observed, the adoption of the solution should avoid unintended risks and couldn't jeopardize the process.

The first step taken was to prepare the organization for AI. Then process operators and engineers at the food firm were trained with the support of professionals from the LCP vendor on the platform. Meanwhile, the industrial process experts collaborate with the data experts at the ML platform to clarify doubts and specific topics about the characteristic of the process to improve.

Regarding OT/IT infrastructure the firm has a robust architecture integrated according to ISA95 standards. This infrastructure generates data from the production processes and is stored in historian-type databases. Then the firm has a very valuable quantity of process data to feed the model offline.

To forecast and optimize the boiler efficiency, the operations team need to predict the optimal biases in fuel intake between the boilers. The operations team used the LCP AI Platform's pre-built machine-learning model templates to configure two models:

- Boiler simulator model: to predict performance of individual assets to predict fuel usage and cost;
- Optimal control parameters model: to identify the optimal control parameters that would minimize the total fuel consumption of all boilers.

Using the LCP AI platform, the operations team uploaded 18 months of time cycle data generated by its gas turbines and boilers, and ambient variables such as ambient temperature, humidity, wind speed, and wind direction. The equipment parameters that were considered were boiler pressure, boiler air flow, boiler fuel input, and boiler air input temperature, among others.

Through visualization and contextualization tools included in the LCP to ease the data discovery process, the operations team observed missing data, determine the importance of characteristics and relationship correlation, and outliers in the process. The missing data in this case did not exceed 10%, which ease the creation of the models.

The next step according to the Lean Startup methodology used, consisted in the evaluation and validation of the model produced in the first stage.

A data exploration and evaluation of data models by the firm Process Managers was performed and analytics experts from the software startup always provided support. The process engineers went thought the models, evaluated different scenarios, and run multiple experiments. A number of methodologies were applied to the entire data set:

1. Data ingestion;
2. Analysis of the correlation between features;
3. Analysis of the feature importance;
4. Visualization of important features.

The platform trained the AI models with the cleaned offline historical data. The LCP automatically selects the best model by cross validating the historical data. The LCP AI was used to configure a simulator model for each boiler to predict the expected fuel usage given the plant steam demand, as well as the operating conditions of the turbine and boiler. An optimizer model was then configured on top of these boiler simulator models to identify the optimal control parameters that would minimize the total fuel consumption of all boilers while respecting the physical and safety constraints of each boiler. The operations team configured a third simulation model, fuel consumption model: to predict how different parameters affect fuel consumption. The top features that affect fuel consumption were valve position(s), water supply temperature to each boiler, and ambient temperature.

The second phase of the methodology was completed with the model going live at the platform online hosted at Azure Microsoft Cloud [26]. Real time from the IIoT solution started to be ingested to the cloud-based solution. All the integration was made by software specialists from the startup. Besides the software solution has an application process interface (API), that simplifies the integration with the IIoT platform. This way the plant users accessed to the ML solution and had continuous support from the software provider.

During the third iteration, according to Lean Startup methodology, the production staff recognized the value that the use case contributed. Results were analyzed by the software supplier and process engineers. The use case was reviewed on a platform mapping session where both teams gained deeper insights into data, business value, and complexity of previously identified use cases.

Once the AI models are productionized, the operations team used the LCP AI to refine the protocols and apply rules to the company's specific operating environment. In the testing phase, there was a 2.5% improvement in thermal performance over previous reports.

LCP AI's automated model training infrastructure ensures that the AI models continuously improve with new data from the steam generation process.

Finally, all the staff of the process team were trained, the platform was adopted, and the process was changed to follow the notifications of the platform. The third phase took other 2 weeks. Figure 3 illustrate the Lean Startup cycle applied to the LCP implementation.

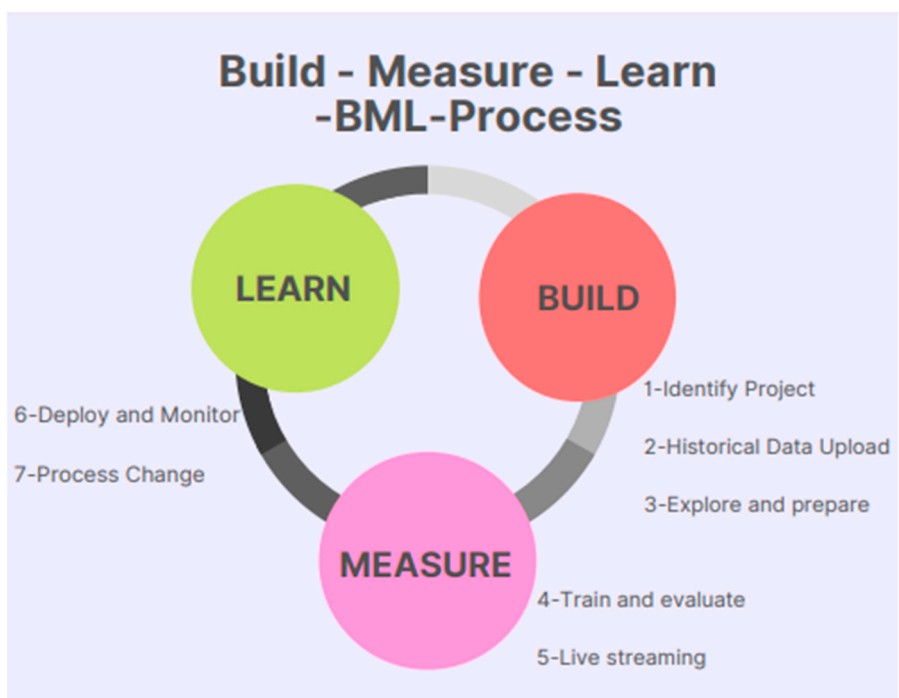

**Figure 3.** Lean startup process structured as Built–Measure–Learn (BML), and the seven. steps followed in the case. Source: Authors.

## 5. Results and Discussion

The project was a good opportunity to demonstrate the LCP ability to provide operationally meaningful interpretation of AI/ML results, and analyze constraints and weaknesses of the methodology.

The first issue to observe is that the LCP produced models able to forecast energy demand from the industrial process faster without the complexity of traditional methodologies that starts from scratch with high coding demand and problem resolution dependent on software experts. In this case the software platform through prebuilt templates provided analysis and visualization to enable software and industrial process experts to identify features that have the biggest impact on the boiler set point temperature. Models prediction can be used to control minimum energy output and cost-saving.

During the analysis between both parties' experts the models produced in the LCP showed correlation between valve opening and energy demand. Historical vs predictive data visualizations within platform showed clear excess energy pushed out. Models' prediction can be used to control minimum energy output and cost-saving.

The platform enabled analysis and visualization of Actual (historical) vs Predicted (optimized) supply temperature showing excess energy used for the past year. By utilizing the Al platform's predictive models, the food company optimized multiple gas turbines, lowering fuel costs by 4%, with the carry-on impact of reducing greenhouse-gas emissions by more than 10 million pounds of $CO_2$ per year. Not only has this contributed to the

company meeting its overall sustainability targets, but it has also helped the plant to significantly reduce energy costs.

The co creating methodology ease the data preparation process, reducing time from data preparation to model training. This is due in part, to early involvement of industrial process people, and the use of the platform automated data preparation and AI building model capabilities. The empowerment of the internal workforce which include process, maintenance, and electrical engineers, seems to be a key issue to succeed, since they are domain experts and understand the plant challenges better than external data scientists.

The continuous improvement team and the operations team gained a deeper knowledge of the process, some process optimization opportunities were highlighted. As an example, the experts were convinced that further gains could be achieved if the model generated forecasts within shorter time intervals. However, this option has not been adopted yet, due to infrastructure limitations. And this is one of the constraints of the platform architecture.

The project also shows some limitations due to data quality. In the studied case the firm had significant amount of good quality data available for Boiler Set Point Temperature optimization. A second case wanted to be approached, but has to be paused due to the lack of important features (features/tags) currently not collected by the IIoT platform.

## 6. Conclusions

The case seems to help clarify how AI solutions could make a traditional factory more efficient. Implementing the LCP through an agile methodology shortens the implementation times, easing the model's co-creation between the industrial and software people.

The implementation methodology and the LCP also impact achieving one of the I5.0 pillars, the centrality of people. The process engineers were involved from the first time, and then they understood the solution and its benefits, achieving a solid compromise with the improvement project. The final solution consists of a human in a loop system, where the operator observes predictions and changes the process, accordingly, attaining AI-driven automation.

The application of AI played a central role in improving the company's operations, helping it accomplish significant sustainability objectives it had identified. This was possible by preparing the organization for AI, committing people, processes, and technology.

The co-creation methodology proposed by using the LCP and the Lean startup approach, seemed to be appropriate to compromise process people in the use of the new tool and ease the cultural change traditional firms need to go through to achieve sustainable goals. On the other hand, software experts who master AI tools could understand the industrial problem domain faster and with the required depth working close to plant experts.

A point to consider is the need for integrated architecture. This could be a weakness for many traditional industries. The case was developed in a plant with a mature architecture that eased the evolution toward using AI. This is a strength for this company that is not common in many other firms. One of the points to consider in this line is the existence of more than 18 months of data generated and managed by the IIoT infrastructure. The strength of this platform can be seen in a low percentage of missing values in the historical data.

Something to observe is the integration with a third-party cloud solution that facilitates AI implementation, providing the architecture needed for high computational resource usage and cybersecurity tools. The LCP works in a reliable multitenant architecture.

I5.0 presents a model for the next level of industrialization, advocating for intelligent supply chains and hyper customization. Integrating data and AI in industrial decision-making is at the core of this new vision, providing the basis for a cognitive factory. This vision, however, is weakened in many industries by missing coordination between the operation and technology domains. Integrating data and AI in industrial decision-making is central in this new vision. To reach full integration, the OT/IT convergence problem should be addressed as early as possible. This way a solid IIoT platforms seems to be the basement to introduce advanced AI tools.

Future research lines intend to go deeper into the I5.0 model to analyze solutions that ease the implementation of AI in industrial processes to gain supply chain resilience.

**Author Contributions:** Conceptualization, methodology, formal analysis, investigation, and writing—review and editing, F.W.M.; data curation, G.P.; writing—review and editing, J.E.T.; Resources, visualization, supervision, project administration, and funding acquisition, A.R. All authors have read and agreed to the published version of the manuscript.

**Funding:** The projects are "Metodologías de abordaje al modelo Industria 4.0 en PyMEs, el rol de la Empresas de Base Tecnológica (EBT), los recursos humanos, y el ecosistema de Innovación", approved by Resolution (R) No. 183-21-UNAJ INVESTIGA Program; and "Mejora de Procesos, Optimización y Data Analytics: Aplicación en Procesos e Industrias de Interés Regional Mediante Estudios de Casos Reales", approved by a resolution of the UNLZ Engineering Faculty.

**Data Availability Statement:** Not applicable.

**Acknowledgments:** The authors wish to thank two research projects that made possible the content of this article.

**Conflicts of Interest:** The authors declare no conflict of interest.

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
