# Peer review of "Adoption Case of IIoT and Machine Learning to Improve Energy Consumption at a Process Manufacturing Firm, under Industry 5.0 Model"

_2504-2289, doi:10.3390/bdcc7010042_

Round 1

Reviewer 1 Report

This is an interesting work that describes a case study implemented in a  food industry where authors applied a low-code AI platform was adopted to improve the efficiency and lower environmental footprint impact of its operations. The paper describes the adoption process of the solution 17 integrated with an IIoT architecture that generates data to achieve process optimization. The case 18 shows how a low-code AI platform can ease energy efficiency, considering people in the process, 19 empowering them and giving a central role in the improvement opportunity. 

Although this paper uses IIoT architecture and performs optimization and implements low-code AI techniques, I suggest few improvements for the manuscript. Though methods are clear, results generated for optimizer model may be described in more detail. Different parameters that affect the fuel consumption may be described in the results section and can be discussed. A chart of these parameters would be more informative and a comparison with previous techniques used for this kind of problem would be added value to the manuscript. In addition, more information about LCP platform protocols and implementation process is required in the manuscript. 

Author Response

I have revised the results and they are described in more detail. Besides the  parameters that affect the fuel consumption are described in the results section and can be discussed.

Finally more information about LCP platform protocols and implementation process are presented

Reviewer 2 Report

The work describes a low-code AI implementation for food industry. It lacks on theoretical background, and does not show quantitative results obtained with such implementation.

I suggest to add more literature analysis, theoretical study and experiments showing the good performances of the proposed work in order to be better evaluated.

Author Response

More literature analysis was added regarding IIoT and AI/ML, and low code platforms cases in traditional industries 

Reviewer 3 Report

Energy consumption and lower environmental footprint impact of industries operations is challenging task that occupies many researchers. Your paper faces a significant problem in industry.

In order to improve this work:

·         In the first section, you should present the task more detailed. As example in line is referred some other studies that are not analyzed.

·         In section 2, you can expand your literature review with other similar studies

·         In section 3, you could describe with more details your approach:

·         Low-Code software (details, some exports for its operation, presentation of its tools)

·         Lean Startup methodology (a short description)

·         In section 4, there is a description of procedure however a presentation of historical data (18 months) is important if possible

·         In section 5, the results are not described detailed some diagrams may stress the results

·         In section 6, you may stress the novelty of this work and underline its weaknesses

Some figures of data and results are going to improve this work.

Author Response

Dear reviewer, all sugestion were addressed.

 In the first section, you should present the task more detailed. As example in line is referred some other studies that are not analyzed.

  • In section 2, you can expand your literature review with other similar studies
  • In section 3, you could describe with more details your approach:
  • Low-Code software (details, some exports for its operation, presentation of its tools), more detail was added
  • Lean Startup methodology  is better described
  • In section 4, there is a description of procedure however a presentation of historical data (18 months) is important if possible. major parameters were explained
  • In section 5, the results are describnbed with more detail

Round 2

Reviewer 1 Report

Authors have explained the top features viz, Value Positions, water supply temperature and ambient temperature. While they described that 42 features were used. A table describing the same would be of interest to the reader.

Results and discussion is not descriptive and needs to be more elaborative for better understanding of the manuscript.

Author Response

Dear reviewer, 

We´ll be working  to answer to your suggetions. First of all we include a figure to make more clear the way the LCP works and ease the adoption of the solution to industrial people.

We add more information about features that were considered in the model, but we didn´t use a table because we couldn´t found how to make it clear to the reader.

Finally we improve the results discussion, and conclusions items.

Hope you find the article suitable to be published.

Best Regards

Federico Walas Mateo

Reviewer 3 Report

In order to improve this work:

·         In the first section, you should present the task more detailed. As example in line is referred some other studies that are not analyzed.

·         In section 2, you can expand your literature review with other similar studies

·         In section 3, you could describe with more details your approach:

·         Low-Code software (details, some exports for its operation, presentation of its tools)

·         In section 4, there is a description of procedure however a presentation of historical data (18 months) is important if possible

·         In section 5, the results are not described detailed some diagrams may stress the results

·         In section 6, you may stress the novelty of this work and underline its weaknesses

Some figures of data and results are going to improve this work.

Author Response

Dear reviewer, 

We´ll be working  to answer to your suggetions. First of all we include more references in the introduction section and complete the idea of what the article is about.

We include more information about cases in other industries,

Besides more information about Lean startup methodology and a figure to make more clear the way the LCP works and ease the adoption of the solution to industrial people were included.

We add more information about features that were considered in the model, but we didn´t use a table because we couldn´t found how to make it clear to the reader.

Finally we improve the results discussion, and conclusions items.

Hope you find the article suitable to be published.

Best Regards

Federico Walas Mateo

Round 3

Reviewer 1 Report

Authors have made immense efforts to make necessary changes to the manuscript. The modifications suggested by the reviewers are addressed. The paper describes the framework of the adoption of Artificial Intelligence and Machine Learning in a traditional industrial environment towards a smart manufacturing approach

Performance evaluation with other similar works may be addressed here to add more on to the authors approach.

Scientific writing may be improved by rephrasing the sentences viz., "Afterward, its results are discussed, to end with the conclusions." in line 86.

Author Response

Dear reviewer, 

We´have been correcting the article to address your suggetions and improve it.

Hope you find the article suitable to be published. Thanks for your suggestions to help producing a better paper.

Best Regards

Federico Walas Mateo
